# A Study on the Pathogenesis of Vascular Cognitive Impairment and Dementia: The Chronic Cerebral Hypoperfusion Hypothesis

**DOI:** 10.3390/jcm11164742

**Published:** 2022-08-14

**Authors:** Weiwei Yu, Yao Li, Jun Hu, Jun Wu, Yining Huang

**Affiliations:** 1Department of Neurology, Peking University Shenzhen Hospital, 1120 Lianhua Road, Futian District, Shenzhen 518036, China; 2Department of Neurology, Peking University First Hospital, 8 Xishiku Street Xicheng District, Beijing 100034, China

**Keywords:** vascular cognitive impairment and dementia, cerebral blood flow regulation, chronic cerebral hypoperfusion, β-amyloid, neuroinflammation, oxidative stress, trophic uncoupling, blood-brain barrier, white matter lesions

## Abstract

The pathogenic mechanisms underlying vascular cognitive impairment and dementia (VCID) remain controversial due to the heterogeneity of vascular causes and complexity of disease neuropathology. However, one common feature shared among all these vascular causes is cerebral blood flow (CBF) dysregulation, and chronic cerebral hypoperfusion (CCH) is the universal consequence of CBF dysregulation, which subsequently results in an insufficient blood supply to the brain, ultimately contributing to VCID. The purpose of this comprehensive review is to emphasize the important contributions of CCH to VCID and illustrate the current findings about the mechanisms involved in CCH-induced VCID pathological changes. Specifically, evidence is mainly provided to support the molecular mechanisms, including Aβ accumulation, inflammation, oxidative stress, blood-brain barrier (BBB) disruption, trophic uncoupling and white matter lesions (WMLs). Notably, there are close interactions among these multiple mechanisms, and further research is necessary to elucidate the hitherto unsolved questions regarding these interactions. An enhanced understanding of the pathological features in preclinical models could provide a theoretical basis, ultimately achieving the shift from treatment to prevention.

## 1. Introduction

Vascular cognitive impairment and dementia (VCID), generally caused by various cerebrovascular diseases (CVDs), is widely recognized as the most common cause of cognitive dysfunction after to Alzheimer’s disease (AD) [1], accounting for at least 20% of dementia cases [2]. VCID is an umbrella term that refers to a heterogeneous group of cognitive disorders ranging from vascular cognitive impairment (VCI) to vascular dementia (VaD) [3]. The risk of developing VCID increases with age, doubling every ~5.3 years [4]. Therefore, as the population ages, the prevalence of VCID is exponentially increased by one in three in men and one in two in women aged 65 years [5]. Within CVD, the most common cause of VCID is likely cerebral small vessel disease (cSVD), which can involve small arteries, arterioles, venules and capillaries in the brain, and ultimately result in arteriolar occlusion, lacunes and white matter changes. Due to the range of these pathological features, VCID shows a variety of clinical manifestations, mainly including pure motor, sensorimotor, pure sensory, ataxic hemiparesis or gait impairment, dysarthria, cognitive dysexecutive slowing and depression, which leads to a great diagnostic challenge for VCID [6]. Additionally, a lot of epidemiological evidence indicates that VCID can develop either alone or in combination with AD. The coexistence of VCID and AD is called mixed dementia, which is more prevalent than pure AD or VCID, accounting for 50% of the dementia in individuals. Studies have shown that vascular comorbidity can be observed in 30–60% of AD individuals, while AD pathologies also exist in 40–80% of VCID cases [7]. Nevertheless, people diagnosed with VCID carry a higher risk of disability and death than those suffering from AD or mixed dementia [8], which might be related to the poor diagnosis and insufficient management of global or localized vascular risk factors, mainly including but not limited to cardiovascular diseases, diabetes mellitus, hypertension, hypercholesterolemia and, in particular, age [9].Thus, VCID is deemed an urgent public health crisis due to its incidence and severity. Fortunately, it is estimated that the sufficient management of vascular disorders can contribute to a 50% reduction in dementia prevalence [10].

The pathogenetic mechanism of VCID has evolved considerably over the years. For many decades, VCID was attributed to multiple strokes caused by vascular risk factors, which provided the credibility that preventing CVD could simultaneously prevent dementia. However, neuroimaging-assisted research was used to determine that diffuse white matter lesions (WMLs) were more common than multiple strokes and were closely related to cognitive impairment [2]. Therefore, the research demonstrated that there must be other pathogeneses of VCID. Impaired cerebral blood flow (CBF) regulation has been proposed and gradually perceived as the dominant presumed mechanism underpinning VCID instead of stroke [11]. Any CVD leading to CBF dysregulation may promote the occurrence and development of VCID. Owing to the distinct vascular causes, VCID possesses a huge neuropathologic complexity in the aged brain, which remains challenging for integrating a global neuropathologic criteria for VCID [12]. Thus, it is extremely significant to determine the exact mechanisms underlying VCID pathogenesis, further developing the harmonized neuropathologic definition. Previous studies have shown that the various pathologic changes induced by chronic, decreased CBF in CVD mainly include arteriolosclerosis, infarcts, microbleeds and WMLs [4]. Thus, it is believed that CBF dysfunction plays an important role in the onset of VCID. In this review, we systematically outline the impact of chronic, decreased CBF on cerebrovascular function and the mechanisms underlying CBF-mediated vasculopathy to VCID, helping us to understand VCID pathogenesis more comprehensively and thoroughly.

In the present review, we summarize the updates on the biochemical basis of VCID according to the recent studies, mainly including the research progress, current limitations and future perspectives. We use a focused search of PubMed/Medline from January 2000 to January 2022, and the relevant literature on VCID is critically reviewed. The following keywords were selected during the search: “chronic cerebral hypoperfusion”, “vascular cognitive impairment and dementia”, “dementia”, “cognitive impairment”, “vascular dementia”, “definitions and classification”, “diagnosis”, “interventions”, “cerebral blood flow”, “animal models” and “biochemistry”. The selected publications in English are subsequently evaluated and included if they are thought to be compatible with the focus of this review. Simultaneously, we exclude the duplicated entries, retracted publications and any other papers on other diseases different to VCID or its subtypes. Additionally, the studies listed in the references are also reviewed in search of more data.

## 2. CBF Regulation and VCID

Cerebral vasculature consists of two blood supply systems: the internal carotid artery (ICA) system conveying approximately 70% CBF to brain, and the vertebral artery system only accounting for 30% CBF, both of which converge to constitute the Willis circle. Extensive branches extending from the Willis circle form a dense cerebro-microvascular network before penetrating the cortex, which plays a critical role in delivering essential nutrients and oxygen to the brain for metabolism [13]. Notably, although constituting merely 2% of the overall body weight, the brain holds approximately 12% circulation of cardiac output and consumes about 25% of glucose and oxygen [14]. Thus, the brain demands a continuous and well-regulated blood supply due to its high energy consumption and shortage of fuel storages, and an even moderate impairment of CBF might have irreparable effects on brain function [15]. In the brain, each neuron is supplied by a corresponding brain blood vessel, which is defined as a neurovascular unit (NVU). An NVU is a particular concept that highlights the complicated interactions among the various cell types in the brain, mainly including vascular cells (endothelium, vascular smooth muscle cells (VSMCs) and pericytes), glial cells (astrocytes, microglia and oligodendroglia) and neurons [16], which plays an important role in CBF regulation, finally maintaining the normal perfusion to the brain and satisfying the neuronal activation with a sufficient delivery of oxygen and nutrients [17]. Therefore, it is important to sustain normal CBF regulation through the dense cerebro-microvascular network that is essential for preserving the NVU function.

Despite facing the changes in the central arterial pressure, the integrated processes still contribute to the relatively constant CBF and microvascular pressure, which is defined as CBF regulation [18]. CBF regulation is extremely complicated with diverse overlapping regulatory patterns for the purpose of complying with numerous unique requirements, such as a high oxygen demand for brain metabolism, no adequate energy stored in neurons, rapidly changing metabolic demands with neuronal activation and a limited space in the closed cranium [11]. This unique function is mainly achieved via the high sensitivity of cerebral vasculature to carbon dioxide [19] and the contractile properties of VSMCs as well as pericytes [20]. Thus, any pathologic changes that can cause metabolic and contractile abnormalities result in CBF dysregulation, mainly including vascular structural lesions due to artery stenosis or occlusion, cerebral hemodynamic alterations, reduced cardiac output and increased blood viscosity owing to the changes in blood components [21]. Additionally, atherosclerosis, hypertension, diabetes, heart diseases and obstructive sleep apnea hypopnea syndrome have also been identified as major risk factors for CBF dysregulation [22]. What are the consequences of CBF dysfunction as a result of these vascular risk factors? The sequence of events mainly include endothelial dysfunction, increases in vascular tone, inward vascular remodeling and vascular hypertrophy. All of these functional and structural changes collectively lead to hypoperfusion at rest [23]. Therefore, chronic cerebral hypoperfusion (CCH) is acknowledged as the chief cerebral hemodynamic alteration of CBF dysregulation, which may lead to ischemic neuronal damage and varied degrees of cognitive deficits, such as VCID [24]. However, the precise pathophysiological mechanisms of CCH underlying VCID still remain a mystery. Thus, we attempted to review the advanced mechanism of CCH-induced brain pathologies in VCID, which enhance the recognition of VCID pathogenesis and may provide the potential therapeutic strategies for VCID via shifting from treatment to prevention.

## 3. Experimental Animal Models of CCH

CCH animal models are considered as good methods to investigate vascular-related dementia. However, owing to the fact that VCID can result from various complicated vascular lesions, it is difficult to replicate all aspects of VCID via a single animal model. Even so, CCH still plays an important role in narrowing the translational gap between animal experiment models and human VCID. Therefore, CCH animal models are valuable preclinical tools for teasing out the specific questions that are difficult to readily address in human studies [25,26]. Due to possessing a satisfactory survival rate and the reproducible behavioral experiment, rats are generally selected to be a constantly applied species for the CCH model through bilateral common carotid artery (CCA) occlusion (BCAO) [27]. BCAO surgery is relatively easy to perform and it merely takes approximately 10 min to ligate both CCAs with sutures. In BCAO rats, permanent vessel occlusion leads to global cerebral hypoperfusion due to the fact that the integral Willis circle can provide incessant but decreased CBF. BCAO-operated rats manifest significant metabolic changes, neuropathological alterations and cognitive impairment with few severe neuronal damages, such as obvious motor dysfunction or seizures [28], which coincides with those observed in VCID patients [29]. However, the BCAO model can result in inevitable damage to the visual pathway via the occlusion of the ophthalmic artery, which potentially influences behavioral evaluations [30]. Additionally, the number of infarcts in BCAO rats is also extremely unpredictable. Fortunately, the four-vessel occlusion (4VO) model with both the CCAs and vertebral arteries blocked is characterized by the predictable ischemic damage and a low rate of seizures [31]. Nevertheless, one remaining problem is that genetic research cannot be performed due to the limited accessibility to molecular technology for rats.

To overcome the drawbacks in the rat CCH models, CCH mice models were established via bilateral CCA stenosis (BCAS) using newly designed microcoils [32]. BCAS can effectively overcome the inherent limitations in rats, including optic nerve injury and the troubles in genetic studies, and easily modify the severity of reduced CBF through adjusting the internal diameter of the microcoils [33]. Similarly, both WMLs and hippocampal atrophy can also be detected in the BCAS mice without optic nerve injury [34]. However, the cerebral infarction in the gray matter in VCID patients is not consistently replicated in the BCAS mice model [9,35]. Moreover, BCAS causes a relatively abrupt CBF reduction that is not consistent with the pattern of gradual CBF reduction in human VCID. Thus, asymmetric CCA surgery (ACAS) was developed. An ameroid constrictor with a 0.5 mm inner diameter was implanted on the right CCA to achieve a gradual vascular occlusion similar to a human’s, and a microcoil with a 0.18 mm inner diameter was simultaneously attached to the left CCA to result in arterial stenosis [36]. Extensive studies have shown that both a gradually decreasing CBF and motor dysfunctions can be observed in the ACAS mice model, except for common WMLs [37]. Therefore, this mice model more closely mimicked the natural history of CVD pathology and accurately exhibited the predominant WML-related cognitive impairment in human VCID accompanied by infarctions and motor deficits, which possess an advantage in exploring feasible pathophysiology.

Finally, several transgenic CCH mice models characterized by cerebral amyloid angiopathy (CAA) were also established. Apolipoprotein E (*ApoE^−/−^*) mice exhibited decreased CBF, BBB disruption, inflammatory cytokine generation, gliosis and spatial memory impairments [38,39]. AD mice models engineered by driving amyloid precursor protein (APP) overexpression were also revealed to be suitable models for CAA. For example, *APPDutch* mice, bearing an APPE693Q mutation, resulted in CAA, strokes and even dementia, and APP23xAPPDutch mice with an *APP KM670/6771NL* mutation overexpressed a seven-fold quantity of mutant human APP and manifested a high Aβ40/42 ratio leading to severe CAA, which indicated that Aβ40 was the major form deposited in the vasculature [40]. Finally, *Tg-SwDI* mice models exhibited a progressively increased Aβ deposition in the cerebrovascular area starting at six months of age, ultimately leading to neuroinflammation, oxidative stress, activated astrocytes and microglia and cognitive impairments [41]. CAA animal models are excellent for studying VCID and provide a better understanding of how Aβ deposition in the vessels can lead to cognitive impairment.

## 4. The Multiple Mechanisms of CCH-Induced VCID

### 4.1. CCH-Induced Aβ Accumulation in VCID

Aβ is a self-aggregating polypeptide with 40–42 amino acids, and it is generated by the proteolytic processing of APP, a ubiquitous glycoprotein generally expressed throughout the brain [42]. Full-length APP, containing 700 amino acids, is a type-I transmembrane protein, which is composed of a large transmembrane N-terminal ectodomain and a short intracellular C-terminal domain, with an N-terminal ectodomain, including the Aβ sequence and C-terminal domain resembling a cell-surface receptor [43]. APP proteolytic processing is achieved by the successive cleavage of secretases, mainly divided into the non-amyloidogenic and amyloidogenic pathway (Figure 1I,II). Aβ is generated from the amyloidogenic pathway (Figure 1II). In the amyloidogenic pathway, β-secretase (the β-site APP-cleaving enzyme 1, *BACE1*) first cleaves APP within the ectodomain to liberate a soluble N-terminal fragment, named soluble APPβ, mainly in the endosomal system from the transmembrane APP holoprotein [44]. The membrane-tethered C-terminal fragment containing 99 amino acids (C99) is subsequently cleaved at the lipid bilayer by the presenilin (PS)-containing γ-secretase multi-subunit complex, generating the extracellular Aβ peptide and the soluble APP intracellular domain (AICD), and AICD is then localized in the nucleus to mediate gene expression [45]. The specific cleavage sites of the APP C-terminal fragment by γ-secretase play an important role in determining the characteristics of the Aβ peptide [46,47]. For example, the longer Aβ species is more aggressive, such as Aβ42, the central production of the neurotoxic Aβ oligomers. Aβ42 tends to deposit in vulnerable brain regions primarily related to learning and memory due to its higher rate of fibrillization and insolubility (Figure 1V). Alternatively, Aβ40 prefers to deposit in non-neuronal cells, such as cerebral vessels (Figure 1VI).

CCH generally results in a protracted period of inadequate amounts of oxygen delivered to the brain. Studies have revealed that hypoxia prominently increases *BACE1* gene transcription and expression through a hypoxia-responsive element (HRE) in its promoter region (Figure 1III). Increased BACE1 can subsequently activate the γ-secretase enzyme to cleave APP, which markedly augments Aβ deposition and neurotic plaque formation, ultimately potentiating neuronal toxicity. Hypoxia-inducible factor 1 (HIF-1), a principal transcriptional regulator for oxygen homeostasis, comprises two basic proteins: an oxygen-regulated expression α subunit (HIF-1α) and a constitutively expressed β subunit (HIF-1β) [48]. As an essential mediator for gene expressions, HIF-1α is constitutively expressed in various cell types, and its half-life is less than 5 min under a normal-oxygen condition on account of being continuously degraded by prolyl-hydroxylase enzymes [49]. However, prolyl-hydroxylase enzymes can be inhibited under hypoxic conditions, which then stabilizes HIF-1α to bind to HRE in promoters or enhancers, activating multiple genes involved in various biological metabolism processes [50]. Previous studies revealed that *BACE1* is one of the downstream targeted genes of this hypoxia signaling pathway, which contributes to an increase in BACE1 expression and activity (Figure 1III) [51]. Zhang et al., highlighted that the levels of BACE1 in the cortex and hippocampus are significantly reduced in HIF-1α conditional knock-out mice, which further indicates that HIF-1α plays an important role in regulating amyloidogenic processing under hypoxic conditions [52]. Additionally, a range of experimental studies have indicated that brain ischemia is also likely to result in Aβ accumulation through impairing the clearance of Aβ (Figure 1IV) [53]. Tesco G et al., revealed that prolonged focal ischemia can reduce BACE1 clearance, and increased BACE1 protein levels can promote Aβ accumulation [54]. On the other hand, neprilysin, a candidate Aβ-degrading peptidase, is also reduced after ischemic insult, which results in defects both in the degradation of exogenously administered Aβ and in the inhibition of endogenous Aβ metabolic levels [55]. However, cerebrovascular alterations mediated by Aβ accumulation can be abrogated via overexpressing the reactive oxygen species (ROS) scavenging enzyme superoxide dismutase, or by making the reduced form of nicotinamide-adenine dinucleotide phosphate (NADPH) oxidase (NOX) subunit NOX2 deficient [56]. In conclusion, CCH plays a critical role in Aβ deposition in the cerebrovascular area and brain parenchyma, and augmenting the oxygen supply via improving cerebral perfusion has potential benefits for preserving cognitive functions.

### 4.2. CCH-Induced Neuroinflammation in VCID

Neuroinflammation primarily characterized by microglial activation and subsequent release of inflammatory factors can eventually result in brain damage, which is revealed to be implicated as a potential connection between CCH and VCID. In a CCH rat model, microglia can be activated from as early as 1 day after BCAO surgery and even remains activated until 28 days [32], which is a unanimous finding in many pre-clinical studies [33,57]. The activated microglia generate multiple pro-inflammatory cytokines, such as interleukin (IL)-1β, IL-6 and tumor necrosis factor (TNF)-α (Figure 2I–III), which provoke an inflammatory response, resulting in the destructive effects on long-term potentiation (LTP) and cognitive function [58]. Additionally, CCH, as an original motivator, can also directly accelerate endothelium activation and alter their phenotypes, making it more porous to pro-inflammatory cells. The activated endothelium releases some pro-inflammatory factors, mainly the intercellular adhesion molecule-1 (ICAM-1) and vascular cell adhesion molecule-1 (VCAM-1), both of which are common adhesion molecules and considered as markers of an activated endothelium [59]. ICAM-1 and VCAM-1 are reclassified as the immunoglobulin super-family and can interact with their corresponding ligands on leukocytes (the integrins CD11/CD18 for ICAM-1, very late antigen-4 (VLA-4) for VCAM-1), and their interactions then result in the adherence and extravasation of leukocytes into brain parenchyma (Figure 2IV). This progress can be accelerated under the CCH condition [60]. Finally, absent in melanoma 2 (AIM2), initiating the formation of the inflammasome complex, is also a key contributor to sterile inflammatory responses. The latest research shows that the AIM2 inflammasome is significantly activated in response to CCH, and the activated AIM2 inflammasome binds to the receptors expressed in the cortex and hippocampus to promote inflammatory responses (Figure 2V), resulting in cellular pathology and cognitive impairment in the BCAS mice model [61]. The genetic deficiency of AIM2 can attenuate inflammasome-mediated proinflammatory cytokine release, apoptosis and cognitive impairment following BCAS [62]. Hence, the inhibition of the AIM2 inflammasome may provide a potential therapeutic target for attenuating cognitive dysfunction in VCID.

Multiple signaling pathways are involved in mediating microglial activation and inflammatory signals, mainly containing mitogen-activated protein kinases (MAPKs), nuclear factor kappa beta (NF-κβ) and triggering receptor expressed in myeloid cells 2 (TREM-2) [63,64,65]. The MAPKs family mainly consists of three groups, namely, extracellular signal-regulated kinase 1/2 (ERK 1/2), c-Jun N-terminal kinase (JNK) and p38, which plays a significant role in regulating the release of proinflammatory cytokines, such as TNF-α and IL-1β (Figure 2I) [66]. The most universal members in the NF-κβ family are p50 and p65, sharing a Rel-homology domain, which are normally discovered in the cytoplasm through the interaction with the inhibitor κB (IκB) family. IκB kinase (IKK) can contribute to the IκB ubiquitination and then dissociate IκB from NF-κβ by phosphorylating IκB inhibitors, ultimately degrading IκB by proteasome. IκB degradation subsequently promotes NF-κβ to translocate into the nucleus and bind to the promoter domains of proinflammatory genes, which eventually triggers the transcription of proinflammatory mediators involved in innate immunity, such as TNF, ICAM-1, IL-6 and iNOS (Figure 2II) [67]. Notably, the factors involved in modulating NF-κB activation can potentially regulate the inflammatory response in ischemic disease. For example, NF-κβ overexpression leads to the death of neurons following middle cerebral artery occlusion, and both p50 inhibitors and p50 knockout in mice models significantly protect the brain from ischemic injury [68]. TREM-2, a vital innate immune receptor, is uniquely expressed on microglial cells. Studies nowadays illustrate that TREM-2 can downregulate TNF-α expression through inhibiting the p38 pathway activation, ultimately alleviating inflammatory responses and immunity (Figure 2III) [69].

As previously outlined in the study, inflammation is a common characteristic in CCH models feathered by microglial and endothelial activation. However, we concluded from clinical observations that there seems to be an association between CNS and peripheral inflammation. It is unclear whether inflammation is an original trigger for VCID and this inflammatory response is provoked by intrinsic or systemic processes. Furthermore, it has been demonstrated that microglial and endothelial activation are more evident in aged mice with BCAS compared to younger mice [70]. Despite much scientific progress in clinical and experimental age-related VCID research in recent decades, our understanding of vascular aging contributing to VCID pathogenesis is still insufficient and various controversies and unsolved questions still remain. Therefore, the underlying mechanism of aging-mediated microglial and endothelial activation in VCID should be fully explored through various CCH models.

### 4.3. CCH-Induced Oxidative Stress in VCID

Oxidative stress can be generally summarized as follows: ROS and reactive nitrogen species (RNSs) are generated beyond normal biological levels and exceed the ability of endogenous antioxidant defense, which results in oxidative and/or nitrosative injury to cellular proteins, lipids and nucleic acids, ultimately disrupting cellular redox signaling processes [71]. Numerous studies have identified that oxidative stress is involved in VCID pathogenesis and is considered as a major contributor to cognitive impairment. NOX family proteins comprise NOX1, NOX2 and NOX4 oxidases, and plays a key role in NADPH oxidase function as the core subunits of the NADPH enzyme. Moreover, NOX family proteins are also considered as the source of ROS in the brain, of which NOX1 is identified as the most primary mediator of ROS generation in the hippocampus and plays a key role in CCH-induced oxidative stress (Figure 2VI). Studies have revealed that NOXs alterations can also significantly change the levels of pro-oxidant and antioxidant enzymes as early as 1 week post 2VO operation, which contributes to cellular damage and cognitive dysfunction [72]. These pathological changes can be alleviated by NOX1 inhibitors [73]. Thus, NOX1 can be elected as a potential target for VCID treatment. Moreover, the subunits of NOXs, including p47phox, p67phox and gp91phox, are also upregulated in 2VO rats to aggravate oxidative damage [74], such as lipid damage that is mainly manifested in the lipid peroxidation products malondialdehyde and 4-hydroxy-2-nonenal, protein oxidative damage reflected by protein modification productions of nitrotyrosine, and DNA oxidation characterized by the production of 8-hydroxy-deoxyguanosine [25].

Increased ROS can downregulate antioxidation-associated transcription factors, such as nuclear factor-erythroid 2-related factor 2 (Nrf2). Nrf2 binds to the antioxidant response element (ARE) to regulate the expression of multiple antioxidants. Studies have revealed that both total and nuclear Nrf2 are significantly downregulated in the hippocampus at two weeks after 2VO surgery, as are the downstream target genes of Nrf2, including NAD(P)H: quinone oxidoreductase 1 (NQO1) and heme oxygenase-1 (HO-1) [75]. These results indicate that CCH can lead to decreased Nrf2 transportation to the nucleus and transcriptional function, and Nrf2 overexpression can assist in reducing cerebral ischemic injury and cognitive impairment in CCH models. On the other hand, increased ROS generated from activated microglia also plays a critical role in promoting endothelial dysfunction via disrupting nitric oxide (NO) signaling [76]. ROS can produce a very potent radical species, peroxynitrite, to sequester NO, which then decreases NO circulation, ultimately accelerating the generation of atherosclerotic plaque. Previous studies have revealed that augmented ROS levels, decreased NO bioavailability and endothelial dysfunction are significantly observed in CCH models. Collectively, CCH-induced oxidative stress potentially damages the functions of the endothelium and accelerate neurovascular dysfunction, which would aggravate inflammation and BBB injury, ultimately leading to WML and cognitive impairment. Accordingly, scavenging free radicals or suppressing inflammation would provide a potential way to alleviate WM injury and cognitive deficits in CCH rodent models.

### 4.4. CCH-Induced Trophic Uncoupling in the NVU

All kinds of cells in the NVU are in a state of close, mutual trophic and metabolic dependences, which ensures the normal physiological processes in the brain. For example, endothelium in the NVU have an ability to promote neuroblast migration along blood vessels and secret abundant brain-derived neurotrophic factor (BDNF) to promote the proliferation and survival of oligodendrocytes and neurons (Figure 3I,II) [77,78,79]. Meanwhile, astrocytes and pericytes are essential for the development and maintenance of endothelial barrier function [80]. Therefore, damage to any type of cells in the NVU would disrupt trophic coupling and also have deleterious effects on the other cell types. Previous studies revealed an impaired neurovascular coupling in the CCH rodent models, and there were interactions between these CCH-mediated pathological alterations, including astrogliosis, significant neuronal loss, inflammation, angiogenesis and neurogenesis, which further indicated NVU remodeling greatly affects various neurological functions [81]. Tropomyosin-receptor Kinase B (TrkB) is an endogenous receptor of BDNF and the combination of these two substances can contact the endothelium and neuroblasts. N-acetylglucosamine oligomers (NAGOs) are generally considered as specific markers of a mature vascular endothelium. Both BDNF/TrkB and NAGO/BDNF double-fluorescent staining can be used to reflect the endothelial neuroprotective support for neurons [82]. J. Shang et al., observed that double-fluorescent intensities of BDNF/TrkB and NAGO/BDNF were significantly reduced in CCH models, indicating that CCH impaired the trophic support for neurons from endothelium (Figure 3I,II) [83]. Additionally, CCH also remarkedly influences the NVU’s functional integrity through impairing the baseline tone in parenchymal arterioles, especially at 28 days post-BCAS surgery [84], in which Adenosine 1 receptor (A1R) plays an important role. A1R activation can trigger the neuron hyperpolarizes in the post-synapse through G-protein coupled inwardly rectifying K^+^ channels [85]. However, previous studies have revealed that A1R is significantly decreased in CCH mice models at 14 days post-surgery (Figure 3III). Meanwhile, adenosine, an inhibitor of neuromodulator, can react with A1R to induce hyperpolarization via the reduction of Ca^2+^ influx and neurotransmitter release, which is also detected in CCH mice models [84].

Furthermore, a large amount of literature has indicated that glial fibrillary acidic protein (GFAP)-positive astrocytes are remarkedly increased within the hippocampus and WM under CCH conditions, which reveals significant reactive astrocytosis that is likely to destroy the connection between astrocytes and vessels [86]. Aquaporin4 (AQP4) is a water channel generally restricted to the astrocytic end-feet processes mainly contacting synapses and vasculature. Thus, end-feet processes play a key role in mediating neurovascular coupling through astrocytic calcium signaling and the release of many vasoactive substances (Figure 3IV). However, AQP4 is mistakenly localized from the vasculature to parenchyma at three months post-BCAS [87]. Both astrocytic activation and subsequent AQP4 mislocalization can induce extensive trophic uncoupling, which then destroys CBF regulation and attenuates the perfusion response of vessels to dynamic demands, ultimately aggravating hypoxia and further promoting various pathological processes. Additionally, AQP4 has an ability to maintain the osmotic balance through eliminating edema induced by some forms of damage (Figure 3IV). The impaired AQP4 polarization of the perivascular end-feet was detected in various ischemic events, which was generally accompanied by BBB impairment [88]. Finally, astrocytes can promote oligodendrogenesis via generating BDNF, which subsequently contributes to the repair of damaged WM [89]. However, this beneficial coupling between astrocytes and oligodendrocytes is disrupted under the CCH condition. Collectively, NVU integrity is likely to have considerable effects on different functions of the brain and no single cell type is likely to be solely responsible for the VCID pathophysiology induced by CCH.

### 4.5. CCH-Induced BBB Breakdown in VCID

Cell-cell interactions between the endothelium and astrocytes form a highly specialized membrane around the cerebrovascular area to underlie the BBB and sustain the functionality of the BBB. The BBB has an ability of inhibiting the entry of hemocytes and plasma components into brain parenchyma and delivering the circulating energy metabolites and essential nutrients to the CNS [90]. Meanwhile, two important characteristics of the microvascular endothelium are involved in maintaining BBB functionality: the continuous tight junctions in the endothelium and the extremely low rate of vesicular transcytosis [91]. Both the destruction of tight junctions and the increase in bulk-flow transcytosis can cause BBB breakdown, promoting the extravasation of multiple circulating macromolecules into brain parenchyma, such as immunoglobulins and albumin, which ultimately leads to the further structural and functional injury of the brain [92]. Neuroma studies demonstrate that BBB impairment can occur in the early stages of CCH, and subsequently precedes the neuroinflammatory reaction and WMLs [93]. In a 2VO rat model, BBB disruption in the paramedian area of the corpus callosum was detected as early as 3 h post-occlusion because of the sudden and severe CBF decrease, and was significantly observed on day 3, but was gradually less prominent from day 7 when CBF began to recover [94]. In BCAS mice models, subtle ultrastructural alterations were observed in BBB as early as 2 h post-surgery, including the opening of tight junctions due to the significantly decreased expression of tight junction proteins claudin-5 and occludin (Figure 4I) [95]. Notably, BBB impairment is more pronounced in regions near WM hyperintensities (WMHs) than the obviously normal WM area in cSVD, which further suggests that BBB impairment plays an important role in associating CCH and WMHs [96]. However, other studies indicate that BBB impairment is not detected until six months after hypoperfusion, although claudin-5 expression is significantly reduced [87]. Thus, the results obtained from the various time points following hypoperfusion indicate that the alterations of BBB integrity might be transient, not sustained.

Matrix metalloproteinases (MMPs), including collagen IV, fibronectin and gelatin, can accelerate the degradation of major extracellular matrix (ECM) constituents and subsequently destroy BBB integrity [97]. Numerous studies have demonstrated that MMPs are consistently overexpressed in activated microglia and endothelium under CCH conditions (Figure 4II), especially MMP2 not MMP-9 [98]. These findings can be verified further through attenuating MMP-2 activity using two strategies: an MMP inhibitor, AG3340, and MMP-2 knockout. Moreover, BBB disruption is proved to account for WMLs, indicating that augmented MMPs may also play a critical role in degrading myelin [99]. All of these pathologic changes mediated by the activation of microglia and the endothelium can subsequently contribute to VCID onset. Additionally, the major facilitator superfamily domain-containing protein 2a (Mfsd2a) is one of the members of the major facilitator superfamily, and plays an important role in the formation and function of the BBB [100]. Mfsd2a, localized in the cytoplasm and plasma membrane, has an ability to sustain an extremely low rate of vesicular transcytosis via inhibiting the generation of caveolae vesicles in the endothelium [101]. However, the level of Mfsd2a expression is significantly reduced in the hippocampus of CCH rat models, which increases the rate of vesicular transcytosis and significant BBB impairment (Figure 4III) [102]. Fortunately, Mfsd2a overexpression in the hippocampus can reduce the rate of vesicular transcytosis and reverse the pathological changes induced by Mfsd2a deficiency. Thus, Mfsd2a-mediated bulk-flow fluid transcytosis plays a key role in regulating BBB permeability in CCH models, which cannot be neglected.

As contractile cells, mature pericytes are localized in the perivascular space to accompany the endothelium and gap junctions in the basement membrane, which play an important role in regulating microcirculatory functions, including CBF regulation, BBB integrity and angiogenesis [103]. Platelet-derived growth factor receptor-beta (PDGFRβ) is a special pericyte marker and plays an essential role in BBB integrity. PDGFRβ-deficient mice are characterized by significantly decreased microvessel length and increased BBB permeability [104]. In CCH models, PDGFRβ expression is obviously reduced (Figure 4IV), which can be restored by administrating a vitamin C/DNA aptamer complex (NXP031) that can increase vitamin C’s antioxidant efficacy [105]. Pericytes preserve BBB integrity mainly through maintaining the two important properties of the microvascular endothelium in the brain. For example, CCH-induced pericytes loss increases endothelial transcytosis and ultimately results in significant BBB impairment [106], which can be reversed by Imatinib, a tyrosine kinase inhibitor, through alleviating the aberrant transforming growth factor-β (TGF-β)/Smad2 signaling activation [93]. On the other hand, pericytes can detach from their perivascular locations under CCH conditions to promotes the capillary permeability of various blood molecules and subsequent neuronal injury. CCH-induced pericyte detachment can also reduce the expression of tight junction proteins, such as occludin, claudin 5 and zonula occludens 1 (Figure 4I) [104]. Although we have learnt a lot about the effects of CCH on pericytes, many questions remain and further research is still necessary to understand the pericyte’s response to CCH.

In conclusion, CCH can promote BBB breakdown via the absence of tight and adherent junctions, the enzymatic degradation of the ECM and an increase in vesicular transcytosis. Abnormal changes in any BBB components under CCH condition lead to BBB destruction. However, further research is necessary to make it clear whether BBB impairment is the initial reason for brain parenchyma damage following CCH or secondary to this damage.

### 4.6. CCH-Induced Demyelination and Failure of Remyelination in VCID

WM, comprising half of the brain and aggregating a large number of nerve fibers, has been gradually recognized to have an equally critical influence on cognition in relation to the cerebral cortex [107]. However, WM merely accepts approximately two-thirds of the blood flow of gray matter, and the blood supply to WM is mainly accomplished by the long penetrating arterioles that arise from the pial cortical network, coincidently in the non-overlapping vascular territories of the anterior and middle cerebral arteries [108], both of which prompt WM to be more susceptible to ischemic injury. Moreover, deep WM is supported by arterioles that directly begin from the Willis circle, and its proximal branches are more vulnerable to mechanical injuries, such as hypertension and arterial stiffness [109]. Neuropathological examinations have revealed that myelin density in WM exhibits a significant attenuation in various forms of dementia, such as VICD, AD and dementia with Lewy bodies, compared to age-matched controls, particularly in VICD [110]. Decreased myelin density predominantly manifests as a hyperintense signal on fluid-attenuated inversion recovery (FLAIR) and T2-weighted magnetic resonance imaging (MRI). Decreased myelin density is the most common and earliest form of brain lesions among the various pathological changes obviously observed in vivo as imaging markers, including microbleeds, microinfarcts, enlarged perivascular spaces and brain atrophy [111]. Furthermore, diffusion tensor imaging and magnetization transfer imaging are thought to be more sensitive to WM tract integrity, and can be used to detect the subtle WM pathologic alternations preceding the visible damage on conventional MRI in VICD humans [112]. Importantly, these sophisticated imaging techniques can also be used as sensitive markers of subtle WM pathology in hypoperfusion mice models [113].

Neuroma studies demonstrate that WMLs can be experimentally induced in various CCH rodent models. Developed WM rarefaction in 2VO rat models was first discovered to be similar to that in humans, which promotes 2VO rat models to be extensively studied [114]. However, pathological alternation in WM occurs very quickly, as early as 1–3 days post-2VO, due to the severe and sudden decrease in CBF, while WM rarefaction via Kluver-Barrera staining in a BCAS mice model avoiding the severe and sudden reduction in CBF was evident from 2 weeks, and the diffuse damage to myelinated axons in WM tracts could be detected at 1 month post-BCAS [33]. The mechanism underlying the degeneration of the myelin sheath induced by CCH can be explained as dying-back gliopathy, in which the injury of oligodendrocytes primarily begins from the most distal processes and then the axon in the myelin sheath, the furthest oligodendrocyte cell bodies [115]. It is known that oligodendrocytes are essential for the formation and maintenance of the myelin sheath because they can generate many growth factors, including insulin-like growth factor 1, and the glial cell-derived neurotrophic factor that contributes to axonal survival [116]. However, oligodendrocytes are very sensitive to hypoxia, and the oxygen level measured by precalibrated sensors in CC of BCAS mice is significantly diminished at 3 days post-surgery, at which time the expression of hypoxia-related genes could also be detected, including HIF-1α, HIF-2α, MMP-7 and neuroglobin, generally considered as compelling evidence for tissue hypoxia [68,117]. Therefore, the considerable loss of oligodendrocytes can occur in various models of CCH. Myelin-forming oligodendrocytes are derived from oligodendrocyte progenitor cells (OPCs) that undergo several stages of development. Reimer et al., reported that the loss of OPCs and mature oligodendrocytes can be simultaneously observed at 3 days post-BCAS, which deprives the trophic support for axons and increases their vulnerability [118]. Additionally, demyelinated axons are exposed to the various deleterious cytokines and free radicals in the hypoxic WM that may destroy axonal energy production and ultimately result in the failure of the Na^+^/K^+^ ATPase. The dysfunction of Na^+^/K^+^ ATPase leads to the accumulation of intracellular Na^+^ and then reverses the operation of the Na^+^/Ca^2+^ exchanger, resulting in intracellular Ca^2+^ accumulation [119]. The pathologic changes in oligodendrocytes poses a considerable threat to axonal integrity causing axonal loss. The myelin-associated glycoprotein (MAG), supporting axon-myelin stability, is generated only in myelinating glial cell bodies and is subsequently transported and expressed only in the adaxonal myelin loop [120]. Therefore, MAG is prone to be more susceptible to ischemic injury than the myelin basic protein (MBP) and proteolipid protein (PLP), and is reduced under the hypoxic condition [121].

In addition to demyelination, CCH can simultaneously cause the failure of axonal remyelination. OPCs are believed to sustain homeostasis in WM and mediate lasting repair in adult brains. As soon as it receives the demyelination signals after injury and disease, OPCs begin to proliferate, migrate and quickly permeate the demyelinated region, and then differentiate into mature oligodendrocytes to form and restore myelin sheaths [122]. Studies have demonstrated that a significant increase in the OPCs’ number can be observed in the brains of VCID patients and CCH rat models [123]. However, the new, immediately produced OPCs in high numbers after ischemia subsequently undergo cell death, because OPCs in the late stage of development are extremely vulnerable to hypoxic injury [124]. Therefore, the regenerative OPCs need to be unable to differentiate into mature oligodendrocytes for remyelination. Fortunately, the phosphodiesterase III inhibitors can prevent OPCs’ death and significantly promote the oligodendrocyte’s maturation, which ultimately restores the WM and cognitive decline [123]. Additionally, Ken et al., revealed that the destruction of trophic support from injured endothelium and astrocytes can also decelerate the vitality of the OPCs pool, and increase the susceptibility to a hypoxic environment in WM [125]. WMLs in various CCH animal models eventually lead to cognitive impairment. Thus, it is necessary to conduct further research to help people understand the earliest events that contribute to WMLs, which could provide more possibilities and opportunities to arrest brain injury at earlier stages and alleviate its impact on cognitive dysfunction and even dementia.

## 5. Conclusions

VCID is the most common contributor to age-associated dementia, secondary to AD, and it is constituted by a heterogeneous group of cognitive impairment factors induced by multiple vascular causes, which makes it difficult to explore the exact pathogenesis of VCID. At first, scholars considered large vascular infarction to be the critical cause of VCID. However, extensive studies indicated that cSVD also contributes to VCID and is even thought of as the most common cause of VCID [126]. Due to the fact that small vessels in the brain are the main contributors of CBF regulation, CCH is universally regarded as a major cerebral hemodynamic alteration in VCID [20]. CCH can subsequently result in a protracted period of ischemia-hypoxia for brain tissue, ultimately inducing the pathological development of VCID. Currently, there might be several biological mechanisms revealed to be involved in the CCH-induced pathological changes in VCID, including neuroinflammation, oxidative stress, trophic uncoupling, BBB disruption and WMLs. Despite the diversity of these underlying mechanisms, there are many interactions among these pathological changes in VCID [2]. NOXs-derived radicals induced by VCID risk factors can further trigger inflammation through activating redox-sensitive proinflammatory transcription factors [127]. In turn, inflammation enhances oxidative stress via increasing the expression of ROS-producing enzymes and receding antioxidant defenses [128]. Moreover, inflammation and oxidative stress, the early events of CCH, then contribute to endothelial dysfunction that mainly contains two kinds of pathogenic manifestations: alterations in BBB permeability and CBF decrease in WM. Contrarily, BBB impairment can also reduce the blood supply to brain tissue and exacerbate brain hypoxia, which further aggravates additional inflammatory response and oxidative stress. Moreover, BBB impairment promotes the extravasation of plasma proteins, including collagen IV, fibronectin and gelatin. The extravasation of plasma proteins is also likely to trigger the inflammatory response from the resident microglia, stimulating the generation of MMP2 and various inflammatory cytokines, such as TNF-α, IL-1β and IL-6 [129]. Inflammation, oxidative stress and BBB impairment subsequently result in trophic uncoupling in the NVU and reduce oligodendrocytes, which has adverse effects on demyelination and remyelination. In turn, demyelinated axons demand more energy support and thus exacerbate brain hypoxia. Collectively, there are indeed strong correlations between these mechanisms underlying VCID, and any kind of pathological mechanism can trigger a vicious circle that perpetuates the mentioned pathogenic processes and accelerates cerebral injury.

AD, the leading cause of dementia, shares common risk factors with CVD, mainly age, hypertension, diabetes mellitus, obesity, hyperlipidemia and hyperhomocysteine [130]. Moreover, numerous studies have found that the functions of the cerebrovascular area are impaired in individuals with early onset AD [131]. However, vascular risk factors are likely to have no obvious correlation with the aggravated burden of AD pathology at death occurring at anold age, while cSVD and cardiovascular disease are associated with this [132]. Therefore, cSVD indeed plays a critical role in the development of AD and VCID, and it is believed that there is a considerable overlap in the pathological alterations between AD and VCID. For example, Aβ deposition can be detected in both VCID and AD with some distinguished points, which still increases the difficulty of clinical diagnosis. The main component of Aβ deposition in VCID is Aβ40 that is located in the cerebral vascular area, while AB42 in AD is mainly distributed in the brain parenchyma. Additionally, both Aβ40 and Aβ42 are remarkedly reduced in the cerebrospinal fluid (CSF) of CAA that is one of the most common causes of VCID, while in the CSF of AD, Aβ42 but not Aβ40 is significantly reduced [133]. Moreover, the pathological evidence of cerebral hypoperfusion was observed in both VCID and AD, mainly inflammation, trophic uncoupling, BBB disruption and WMLs, but with distinctions in the distribution and associated pathologies [7]. MAG and PLP1 are distinctly sensitive to oxygenation changes in tissues and can be used to quantify the ischemic damage. Both MAG and PLP1 are stable for over 72 h under post-mortem conditions due to their slow turnover in vivo. Thus, a reduction in the ratio of MAG: PLP1 of a post-mortem brain reveals cerebral blood perfusion over a period of several months prior to death [134]. The reduced MAG:PLP1 ratio was detected in the cerebral cortex in AD while only being observed in the parietal WM of VCID [135]. Based on the mentioned overlapped lesions, cerebrovascular pathologies as the integral parts in AD play an important role in cognitive impairment, which provides sufficient evidence for the hypothesis that improving vascular function is beneficial for AD treatment. Simultaneously, cognitive impairment is also induced by the combined damage of the vascular and neurodegenerative pathologies because CVD and AD are common in the elderly population. Thus, it is essential to distinguish AD, VCID or mixed dementia according to the special differences among them.

Monogenetic factors can also be involved in the development of VCID in relatively young people. Cerebral autosomal dominant arteriopathy with subcortical infarcts and leukoencephalopathy (CADASIL) is currently the most common hereditary cause of VCID [9]. Patients with CADASIL have no obvious clinical stroke or severe CVD, but they present some evident neuroradiological changes, such as WMLs or leukoaraiosis, which is generally an effect secondary to cSVD [136]. In addition to WMLs, neuroradiology can also be characterized by microbleeds, lacunar infarcts and brain atrophy, which are associated with clinical manifestations in CADASIL patients, including recurrent strokes, cognitive impairment and migraines with aura or mood disturbances [137]. Numerous studies have revealed that nearly all CADASIL individuals are accompanied by the significant missense mutations in the *Notch3* gene that either create or eliminate cysteine residues [138]. In addition to CADASIL, few specific genetic factors related to hereditary small-vessel syndromes have generally not been reported as the causes of VCID. Although the *ApoE* e4 allele is generally considered as a strong susceptible gene for cardiovascular diseases and AD, the significant relationship between *ApoE* e4 and VCID is currently not illustrated [139]. However, studies have found that *ApoE* alleles, including ɛ2 and ɛ4, have remarkable effects on the development and clinical severity of CAA, which suggests that there may be a link between *ApoE* and VCID. Additionally, there are also several candidate genes associated with endothelial NO synthase, oxidative stress and inflammation likely to be involved in VCID. For example, the mutation of TGFβ1 repressor HTRA1 would induce cerebral autosomal recessive arteriopathy with subcortical infarcts and leukoencephalopathy (CARASIL) [140], frameshift deletions in the three-prime repair exonuclease 1 (*TREX1*) gene mediating the onset of autosomal dominant retinal vasculopathy characterized by cerebral leukodystrophy [141], and mutations in the type-IV collagen alpha 1 (*COL4A1*) gene, encoding the COL4A1 chain, are also significantly associated with leukoencephalopathy and intracranial hemorrhage [142]. Although mutations in these candidate genes can eventually result in WMLs, there is still not enough evidence to support the idea that the candidate genes can be regarded as the specific genes for VCID. To our knowledge, there is no publicly available data on the CCH development in mice with deletions of these candidate genes, and whether the deficiency of these candidate genes eventually leads to CCH-induced VCID remains to be confirmed. Therefore, it is necessary to explore the risk gene factors further and provide new, potential modifying therapies for VCID.

Although preclinical and clinical studies on VCID have assisted us to make great advances in our knowledge of VCID, several fundamental questions still remain to be addressed. Firstly, as summarized in this review, various CCH models were used to simulate human VCID, and the essence of these models was to extensively reduce cerebral perfusion. Thus, monitoring CBF appears to be particularly critical in the related studies. However, sensitive methods used in measuring CBF, such as the laser Doppler perfusion image system, are limited to cortical regions, and the inherent CBF in WM is relatively low, which makes it difficult to detect the subtle alterations of blood flow. Thus, it is presumably a great challenge for the currently available technology to access WM, resulting in few studies on the monitoring of CBF alterations in WM. Additionally, the monitoring of CBF alterations is generally performed when animals are anaesthetized, which is likely to be different from that obtained in conscious animals, especially using vasodilatory anesthetics, such as isoflurane. Therefore, it is necessary to develop more sensitive and accurate CBF detection techniques to overcome the mentioned weaknesses. Secondly, in addition to the decreased CBF mediated by various CCH methods, various pathological alterations may be induced by the surgery itself through contributing to special alterations in vasculature, which presents no obvious relationship with the lack of perfusion. For example, surgical operations, whether using microcoils or ligation of vessels, can modify CBF autoregulation, accelerate vascular stiffness and affect CSF dynamics, each of which is considered to be deteriorated for VCID [143]. Moreover, the application of microcoils and constrictor devices increases the localized cellular inflammation and subsequently promotes pathological changes and behavior impairment [87]. Thus, in addition to perfectly recapitulating all the characteristics of VCID, it is important to distinguish whether surgery-related alterations can also occur in the hypoperfusion models, and try to establish a new animal model to reduce or eliminate these pathological changes induced by surgery itself. Thirdly, owing to the lower cost and easier operation, rodent models are the most commonly used to perform the CCH experiments. However, WM in rodents merely occupies an extremely small part in their brains, while occupying approximately a 50% volume of human brains [144]. Alternatively, non-human primates not only have a cerebral architecture similar to humans, but also possess a larger WM volume. Moreover, amyloid deposition and tau pathology can also be detected in the aged brain of non-human primates, which is seldom observed in rodent brains [145]. Additionally, rodent models are limited to achieve a behavioral repertoire in line with human VCID. These inherent defects in rodent models severely impede further studies on the pathological changes in WM structure and function. Thus, a non-human primate model is urgently established to largely represent the best model for exploring extensive WM pathologies and complicated behaviors in VCID. In brief, the effective therapies for VCID are currently absent, and promoting vascular health is probably critical to prevent both vascular dysfunction and neurodegeneration in VCID. Thus, eliminating the gap between CCH rodent models and human VCID through constantly improving animal models will be a great help for us to effectively translate the information obtained from animal models, and explore more effective and potential treatments for VCID.

## Figures and Tables

**Figure 1 jcm-11-04742-f001:**
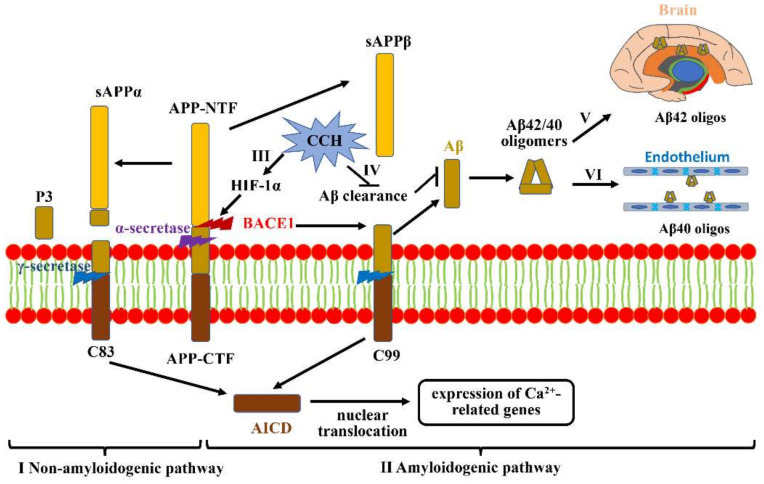
**Aβ accumulation and aggravation under CCH.** (**I**) Non-amyloidogenic pathway: APP is first cleaved by α-secretase to generate sAPPα and C83. C83 then produces P3 and AICD under the action of γ-secretase, which is shifted to the nucleus to regulate Ca^2+^-related gene expression. (**II**) Amyloidogenic pathway: full-length APP can also be cleaved by BACE1 to release soluble APPβ from the cell membrane and retain C99. Subsequent cleavage of C99 by γ-secretase generates Aβ42/40. (**III**) CCH-mediated hypoxia can promote *BACE1* gene expression by upregulating HIF-1α. (**IV**) CCH is also likely to increase Aβ accumulation via impairing Aβ clearance. (**V**) The more aggressive Aβ42 tends to deposit in vulnerable brain regions that are primarily associated with learning and memory. (**VI**) Aβ40 prefers to deposit in non-neuronal cells, mainly including cerebral vessels.

**Figure 2 jcm-11-04742-f002:**
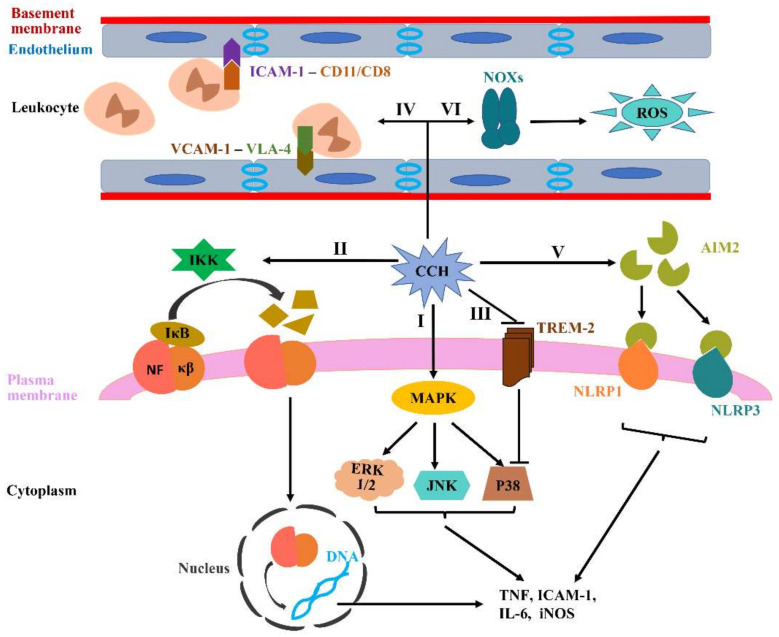
**CCH-induced neuroinflammation and oxidative stress.** (**I**–**III**) The activated microglia by CCH can generate various pro-inflammatory cytokines, such as IL-1β, IL-6 and TNF, in which multiple signaling pathways are involved, including MAPKs, NF-κβ and TREM-2. (**IV**) CCH can directly accelerate endothelium activation. Activated endothelium express more ICAM-1 and VCAM-1 to bind to CD11/CD18 and VLA-4, which contribute to the adherence and extravasation of leukocytes into brain parenchyma. (**V**) AIM2 inflammasome signaling can be activated during CCH, which increases microglial activation and subsequently promotes the generation of pro-inflammatory cytokines. (**VI**) NOXs, recognized as the primary source of ROS in the brain, are thought to be the key contributing factors to CCH-related oxidative stress.

**Figure 3 jcm-11-04742-f003:**
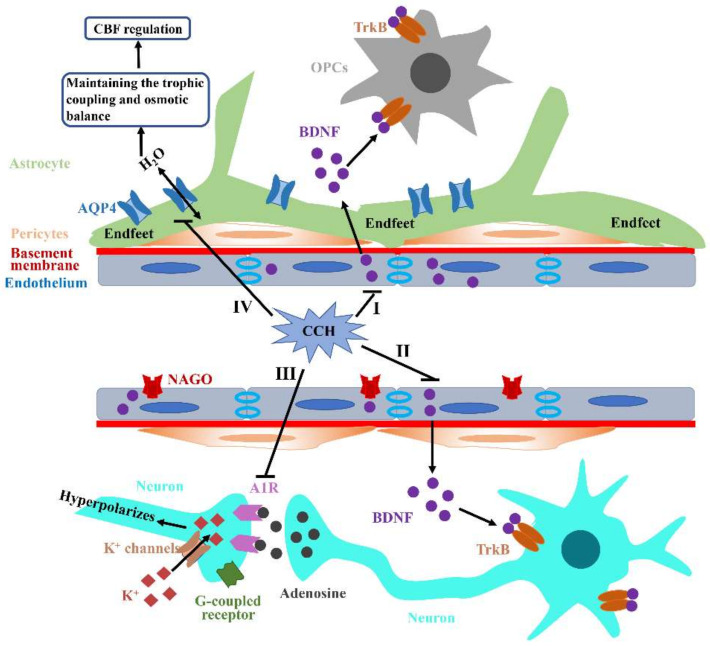
**Aβ accumulation and aggravation under CCH.** (**I**,**II**) The abundant BDNF produced by endothelium in the NVU can bind to TrkB receptors on OPCs and neurons to promote their proliferation, migration and survival, which can be disrupted by CCH. (**III**) CCH can significantly decrease A1R expression and subsequently impede neuron hyperpolarizes in the post-synapse through G-protein coupled inwardly rectifying K^+^ channels. (**IV**) AQP4, as a water channel, is generally restricted to astrocytic end-feet processes that mainly contact synapses and vasculature. Additionally, AQP4 also plays an important role in maintaining osmotic balance through eliminating edema induced by various damages. CCH can lead to AQP4 mislocalization and then induce extensive trophic uncoupling and osmotic imbalance, ultimately destroying CBF regulation and attenuating perfusion to the brain.

**Figure 4 jcm-11-04742-f004:**
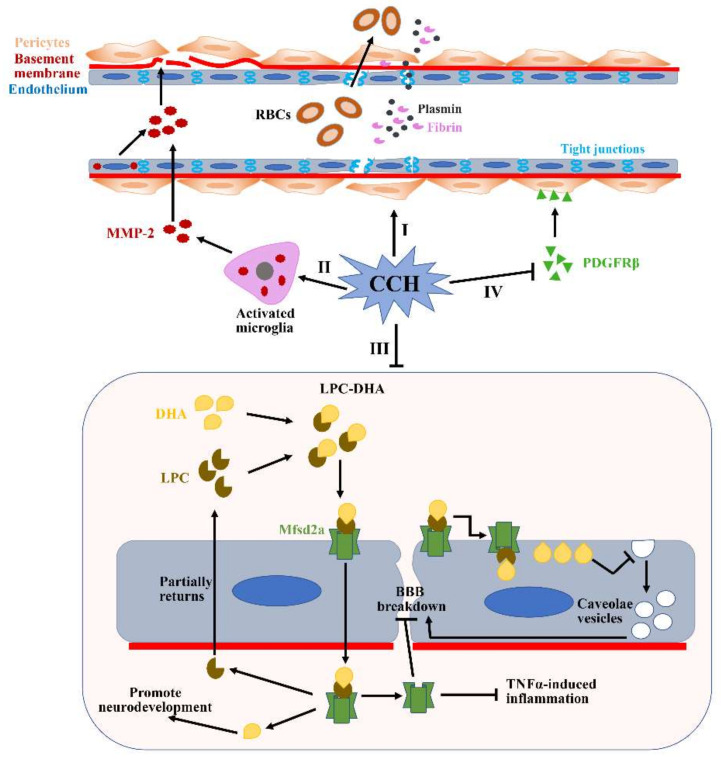
**Potential mechanisms of BBB breakdown during CCH.** (**I**) Pericytes detach from the perivascular positions under the CCH condition, which subsequently reduce the expression of tight junction proteins. Additionally, CCH can also directly disrupt the expressions of tight junction proteins. (**II**) CCH can upregulate levels of MMP2 expression in the activated microglia and endothelium. Increased MMP2 can accelerate the degradation of many major ECM constituents, such as collagen IV, fibronectin and gelatin. (**III**) Mfsd2a plays a critical role in transporting DHA into the cells and maintaining the low rate of vesicular transcytosis. However, the expression of the Mfsd2a protein is significantly reduced during CCH, which causes augmented vesicular transcytosis. (**IV**) CCH-mediated PDGFRβ reduction can result in significantly decreased microvessel lengths. All of these pathological changes induced by CCH can subsequently lead to significant BBB impairment and contribute to the translation of many substances from the blood vessels to the brain parenchyma, such as RBCs, plasmin and fibrin. RBCs: red blood cells; DHA: docosahexaenoic acid; LPC: lysophosphatidylcholine.

## Data Availability

Not applicable.

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
