# Peer review of "A Study on the Pathogenesis of Vascular Cognitive Impairment and Dementia: The Chronic Cerebral Hypoperfusion Hypothesis"

_jcm, 2022, doi:10.3390/jcm11164742_

Round 1

Reviewer 1 Report

The purpose of this manuscript is to emphasize the contributions of chronic cerebral hypoperfusion (CCH) to vascular cognitive impairment and dementia (VCID) and illustrate the current findings about the mechanisms involved in CCH-mediated VCID pathological changes.

·       In the introduction, I suggest to amplify the explanations of VCID (incidence, clinical manifestations, age, sex differences) with the aim of further emphasizing the importance of studying this argument.

·       Furthermore I suggest to include in the introduction a starting point about the eziopathogenetic mechanism of VCID (Which are the conventional interpretations of the eziopathogenesis of VCID?)

·       I suggest, after the introduction, to insert a paragraph on the research methodology of the studies that will be used and discussed during the paper.

·       I suggest you to review the syntax because sentences are often long and repetitive; it is often difficult to understand their meaning.  

·       Because it is difficult to replicate the VCID in animals, I suggest to reduce the paragraph about animals.

·       The figures are very nice; they are very explanatory and help the reader to better understand the text.

Reviewer 2 Report

Major

This review paper claims that CCH is the main cause of VCID, there is not enough evidence.

CCH and CBF dysfunction is ambiguously expressed as the same entity (line 100), but they are different. Although the animal model (paragraph 3 which starts from line 107) of CCH are examples of large artery disease, the CCH in VCID is related to small vessel disease.

The title of paragraph 4 (starts from line 180) is ‘CCH induced’, but the following issues are all ‘CCH-medicated’ which is not the main problem for VCID

Usually, monogenic disease provides clues to the pathogenesis of complex genetic diseases. For example, the basis of the amyloid theory in AD is derived from the functions of PSEN1, PSEN2, and APP genes that cause familial AD. However, in the case of VCID, it seems difficult to find evidence related to CCH in the functions of NOTCH3, HTRA1, TREX1, and COL4A1.

In addition, it is recommended for authors to describe researches against hypothesis in addition to support hypothesis.

Minor

line 48 needs to be supported by references

line 99-102 is not compatible with reference 24. Reference 24 was published in 2005 and mainly dealt with AD, which is not VCID.

line 36 is named -> is named as

line 48 though->thought

ApoE4 -> APOE e4
